# The effectiveness of pharmacological and non-pharmacological interventions for fatigue in people living with chronic kidney disease: A protocol for a systematic review

Hannah Mary Louise Young[1,2,3]*, Lisa Ancliffe[4], Ffion Curtis[5], Oluwafemi Efuntoye[6], Matthew PM Graham-Brown[7], Selina Lock[8], Kathrine Parker[9,10], Chris Rolfe[11], Ellesha Smith[12], Aniebiot-Abasi Udofia[13], James O Burton[1,7], Caroline Booth[14], Ishida Yuko[15], Daniel Scott March[7]

**1** NIHR Leicester Biomedical Research Centre, University of Leicester and University Hospitals of Leicester NHS Trust, United Kingdom, **2** Diabetes Research Centre, College of Life Sciences, University of Leicester, Leicester, United Kingdom, **3** Therapy department, University Hospitals of Leicester NHS Trust, Leicester, United Kingdom, **4** Department of Therapy, Royal Free London NHS Foundation Trust, London, United Kingdom, **5** Liverpool Reviews & Implementation Group (LRiG), University of Liverpool, Liverpool, United Kingdom, **6** Department of Renal Medicine, Queen Elizabeth Hospital Birmingham, Birmingham, United Kingdom, **7** Division of Cardiovascular Sciences, University of Leicester, Leicester, United Kingdom, **8** Library and Learning Services, University of Leicester, Leicester, United Kingdom, **9** Department of Renal medicine, Manchester University NHS Foundation Trust, Manchester, United Kingdom, **10** Division of Pharmacy and Optometry, University of Manchester, Manchester, United Kingdom, **11** Patient and Public Representative and Chairman Exeter & District Kidney Patients Association, Exeter, United Kingdom, **12** Department of Population Health Sciences, University of Leicester, Leicester, United Kingdom, **13** Department of Nephrology, University Hospital of Leicester NHS Trust, Leicester, United Kingdom, **14** Department of Nephrology, Evelina London Children's Hospital, London, United Kingdom, **15** Department of Nutrition and Dietetics, Greater Glasgow and Clyde NHS, United Kingdom

* hy162@leicester.ac.uk

## Abstract

Chronic kidney disease (CKD) affects approximately 850 million people worldwide and is associated with a substantial and growing symptom burden. Fatigue is one of the most common and debilitating symptoms across all stages of CKD, with prevalence far exceeding that of the general population. It profoundly affects quality of life, daily functioning, and clinical outcomes, underscoring the need to identify and evaluate effective pharmacological and non-pharmacological strategies for its management. This protocol outlines methods for a systematic review to evaluate the efficacy, safety, and perceived effectiveness of interventions for fatigue in people with CKD. We will include randomised controlled trials, non-randomised controlled trials, before-and-after studies, and qualitative evaluations in adults and children with any stage of CKD. Searches will be conducted in MEDLINE, Embase, CINAHL, Web of Science Core Collection, PsycINFO, and ClinicalTrials.gov, alongside grey literature via Google Scholar. No language or publication date restrictions will be applied. Study selection will follow a two-stage screening process, with two reviewers independently assessing titles/abstracts and then full texts using predefined eligibility criteria. Data

**Data availability statement:** The current manuscript is a study protocol for a systematic review and meta-analysis which is still currently in progress. No results are available to be reported yet at current stage. All deidentified research data will be made publicly available when the study is completed and published.

**Funding:** This work was supported by the National Institute for Health and Care Research (NIHR) Leicester Biomedical Research Centre. HMLY is supported by a NIHR Advanced Fellowship (NIHR302926). The views expressed in this publication are those of the authors and not necessarily those of the NHS, the NIHR or the Department of Health and Social Care. The funders had no role in the study design; collection, analysis, and interpretation of the data; writing the report; and the decision to submit the report for publication.

**Competing interests:** The authors have declared that no competing interests exist.

will be extracted using a standardised form, capturing study characteristics, interventions, and outcomes. Risk of bias will be assessed with the NIH Study Quality Assessment Tools or the Joanna Briggs Institute Critical Appraisal Checklist for Qualitative Research. Results will be synthesised narratively, and, where appropriate, pooled using meta-analysis. Certainty of the evidence for the primary outcome will be assessed using the GRADE approach and presented in a Summary of Findings table. This review will provide a comprehensive synthesis of evidence on interventions for fatigue in CKD, highlight gaps to guide future research, and inform the UK Kidney Association Symptom Guidelines.

## Introduction

Chronic kidney disease (CKD) affects approximately 850 million people worldwide [1]. The burden of CKD continues to grow, driven largely by global population rises, increasing prevalence of multiple long-term conditions and an aging population [1–3].

Fatigue is one of the most common and challenging symptoms faced by people living with all stages of CKD [4]. Prevalence rates range from 40–94% dependent on the assessment tool used for assessment [4–6]. There is no one universally accepted definition of fatigue in CKD [6,7] Fatigue is a complex, multidimensional phenomenon, with complex causes spanning physical, cognitive, emotional, and social domains [4]. Fatigue significantly impacts quality of life, relationships, work ability, and capacity to engage with treatmentand also associated with increased risk of mortality [4,8–11]. Despite its prevalence and impact, fatigue in CKD is often under-recognised and poorly managed in routine care [11,12].

Previous systematic reviews of fatigue in CKD reveal that a limited range of pharmacological and non-pharmacological approaches to the management of fatigue have been evaluated, within a restricted range of CKD stages [13–15]. A systematic review by Gandra et al. (2010) evaluated the effects of erythropoiesis-stimulating agents (ESAs) in non-dialysis CKD patients with anemia, identifying 11 relevant studies [15]. Eight reported statistically significant improvements in energy, although overall findings were constrained by methodological limitations [15]. A similar systematic review by Johansen et al (2012) which evaluated the impact of ESA on people receiving dialysis concluded that partial correction of anemia with ESA resulted in improvements in fatigue. These results were limited by the reliance upon narrative synthesis [13]. Picarello et al. evaluated the impact of psychosocial interventions targeting fatigue in people with end stage kidney disease (ESKD) [14]. Meta-analyses demonstrated a significant improvement in fatigue following intervention,however, most studies were of poor quality [14]. More recently, a Cochrane systematic review examined a broader range of pharmacological and non-pharmacological interventions for fatigue in people with ESKD receiving dialysis [16]. There was low certainty evidence for the beneficial effects of exercise, aromatherapy, acupressure, massage and uncertain effects for other pharmacological and non-pharmacological interventions [16].

Lack of emphasis on the management of fatigue in routine CKD care has been driven, not only by lack of high-quality trials but also a dearth of clinical guidance on the treatment of fatigue within this population [4,17]. Contemporary CKD guidelines, including those from KDIGO that address fatigue in the dialysis population, emphasise correcting underlying medical contributors such as anaemia and optimising dialysis [18] However, they provide little explicit guidance on managing fatigue through the broader strategies offered by the renal multidisciplinary team. In several other chronic conditions where fatigue is recognised as a core symptom, guidance increasingly adopt a holistic approach that integrate both biomedical and supportive strategies to address the multifaceted impact of fatigue [19–21]. This disparity highlights an urgent need to improve how fatigue is managed within the nephrology setting.

A systematic review is warranted to assess the effectiveness of existing interventions for fatigue across all stages of CKD and consequently inform the development of robust, evidence-based, patient-centred UK Kidney Association clinical guidelines for fatigue management in CKD.. Accordingly, the aim of this review is to:

- evaluate the efficacy and safety of pharmacological and non-pharmacological interventions for the management of fatigue in people living with CKD and;

- explore the perceived effectiveness of these interventions from the perspective of people living with CKD. This will allow us to determine any effects of interventions on fatigue which may be difficult to capture using existing objective measures.

## Materials and methods

This systematic review protocol is reported in accordance with PRISMA-P guidelines (S1 Table) [22]. The review protocol was registered with the International Prospective Register of Systematic Reviews (PROSPERO) on 11.09.2025 (registration number CRD420251102339). In the event of protocol amendments, the date of each amendment will be accompanied by a description of the change and the rationale.

### Eligibility Criteria

Studies will be selected according to the criteria outlined below.

**Study designs.** Studies included will be randomised controlled trials (RCTs), non-randomised controlled trials, and observational studies (before and after studies) that have evaluated an intervention to manage fatigue. Due to the multifactorial nature of fatigue and its wide impact upon the lives of people living with CKD, qualitative evaluations of relevant interventions will also be included. Qualitative studies which focus only on the experience of fatigue, without reference to the effects of an intervention, and which do not focus upon the perspective of people living with CKD, or where these perspectives cannot be isolated from other interest-holders (e.g., in mixed focus groups) will be excluded. Literature reviews, editorials, and commentaries will all be excluded, but their reference lists will be reviewed for potentially relevant publications. Relevant conference abstracts will be included where they provide sufficient data for synthesis. No restriction will be placed on year of publication or minimum follow-up period.

**Participants.** Due to the wide-ranging impacts of fatigue across all CKD stages, we will include all adults and children who are diagnosed with:

(1) CKD stages G3A-G5 (eGFR<60); OR

(2) On dialysis (HD or PD); OR

(3) Kidney transplant recipients; OR

(4) End-stage kidney disease (ESKD) OR

 

(5) Post-dialysis withdrawal OR

(6) Being conservatively managed (full active treatment without KRT)

Studies involving animals will be excluded.

**Interventions.** Included interventions include both pharmacological and non-pharmacological interventions for fatigue management. Given the absence of a universally accepted definition of fatigue, we will adopt an inclusive approach to study selection. Eligible studies will include those that (i) explicitly define fatigue, (ii) employ a recognised fatigue-specific outcome measure, or (iii) report fatigue as a subscale within a broader construct (e.g., health-related quality of life). This approach is intended to capture interventions that, although not explicitly designed to target fatigue, may nonetheless exert secondary effects on it (e.g., exercise-based interventions). By doing so, we aim to enhance the comprehensiveness and relevance of the synthesis.

**Comparators.** For RCTs and non-randomised controlled studies, comparators will include usual care and other interventions, as the latter may contribute to the relative effect estimates within a network meta-analysis. Control groups that receive usual care plus a sham intervention or a placebo (for pharmacological interventions) will also be included.

**Outcomes.** The primary outcome of interest will be the effectiveness of pharmacological and non-pharmacological interventions for reducing fatigue. Fatigue will be using patient-reported outcomes which directly measure fatigue (e.g., the Chalder Fatigue Scale) or include a fatigue subscale as part of a measure assessing a broader construct. Fatigue has been selected as the primary outcome due to the focus of the review, but also because it is consistently identified as one of the most important outcomes for people living with CKD across a range of age groups, disease stages and conditions [23–28].

To address conceptual heterogeneity in fatigue measurement, outcome measures will be prioritised using a predefined hierarchy. Primary focus will be given to validated fatigue-specific outcome measures. Fatigue subscales derived from multidimensional symptom or quality-of-life instruments will be secondary. Where multidimensional outcomes are used, only fatigue-specific subscales will be extracted. Where conceptually justified, different fatigue domains will be analysed separately.

Furthermore, in view of the heterogeneity of fatigue syndrome across chronic kidney disease cohorts, the primary outcome will be reported separately for four groups: patients undergoing dialysis, non-dialysis patients (including those in pre-dialysis stages), kidney transplant recipients, and individuals with CKD receiving conservative management and for adults and children. The primary outcome will likewise be analysed separately according to categories of pharmacological and non-pharmacological interventions.

Where addressing fatigue is the primary focus of the intervention secondary outcomes include:

• Fatigue measured using outcomes which are not patient-reported;

• Health-related quality of life;

• Mental health (e.g., depression and anxiety);

• Sleep quality;

• Withdrawal from treatment;

• Physical function and mobility;

• Cognitive function;

• Life participation;

• All-cause mortality;

• Cardiovascular mortality

Adverse events / safety outcomes of interventions in people living with CKD will also be included, including (but not limited to) worsening of fatigue.

All secondary analyses will also be stratified by CKD stage, as per the primary outcome, and by age group (adult/child).

Recognising the multidimensional nature of fatigue, we will also examine the perspectives of individuals with CKD on how interventions affect their fatigue, as certain impacts may not be fully captured by conventional objective or patient-reported outcome measures.

**Timepoints.** Outcomes will be extracted at all available time points to allow for immediate and long-term effects of interventions to be examined where possible.

**Setting.** All relevant healthcare settings (hospitals, outpatient clinics, dialysis centres, community and home settings, and hospices).

**Language.** Non-English trials or studies deemed relevant at the title and abstract screening stage will be included. Decisions about these reports will be made on a case-by-case basis at the full-text screening stage.

## Information Sources

Literature search strategies were developed by an information specialist, including terms related to CKD, fatigue and relevant study designs. Search terms specifically relating to conservative management were not included as we found these were identified already within the CKD terms. We will search MEDLINE (via Ovid), Embase (Ovid), CINAHL (EbscoHost), Web of Science Core Collection (APA), PsychINFO (EbscoHost) and ClincialTrials.gov (for ongoing and unpublished trials) from the date of inception onwards. The search strategy for MEDLINE is included in S2 Table. Database searches will be supplemented with a Google Scholar search with the first 200 titles screened for inclusion. Searches will be supplemented by reviewing reference lists of key included studies and existing systematic reviews and meta-analyses. We will also explore whether there are qualitative studies linked to relevant quantitative studies and screen them for inclusion in the qualitative synthesis.

## Data Management

Literature search results will be uploaded to the web-based screening and data extraction tool Covidence (Veritas Health Innovation Ltd., Melbourne, Australia). Covidence will be used to facilitate collaboration between reviewers across all stages of the study selection and extraction process.

## Screening procedure

Titles and abstracts of all identified studies will be independently screened by two reviewers in line with the pre-defined inclusion criteria. Full text papers will subsequently be requested and assessed by two reviewers. Any discrepancies between the two reviewers will be resolved by recourse to a third reviewer. We will seek additional information from study authors where necessary to resolve questions about eligibility and record the reasons for exclusion.

## Data extraction

Data extraction will be performed using an adapted Cochrane Data Extraction Template. A single extraction form will be used, but some domains will only be relevant to quantitative and qualitative studies respectively. Qualitative studies will be highlighted and then qualitative findings will be extracted verbatim into NVivo (QSR International Ltd 2022, Version 20) by a single reviewer for subsequent analysis and synthesis. Draft data items to be extracted are outlined within Table 1. Information extracted in relation to equality, diversity and inclusion of the study sample will be guided by PRO-EDI and include items such as age, sex, gender, race, ethnicity, socioeconomic status, level of education, and employment [30]. Data relating to non-pharmacological interventions will be extracted using the TIDIER guidelines as a framework [31].

**Table 1. Data items to be extracted.**

| Study characteristics | **_All studies_**<br>**Date/title, author, year, country, aim, study design, unit of allocation, duration of study, inclusion criteria, exclusion criteria, method of recruitment of participants, duration of intervention, primary outcome, other outcomes, and funding source.**<br>**Qualitative studies:**<br>**Context, stakeholders, theoretical approach, data collection, and analysis methods.**<br>**Study findings will also be extracted, defined as all of the text labelled as 'results' or 'findings' in study reports [29].** |
| --- | --- |
| **Participant characteristics** | Population description, including data relating to equality, diversity and inclusion, CKD stage, renal replacement therapy status, co-morbidities, fatigue characteristics (e.g., type of fatigue, severity of fatigue, duration of fatigue) at baseline, and total number randomised (for trials). |
| **Intervention and Comparators** | Intervention names (pharmacological or non-pharmacological), number randomised to group-sample size, intervention description and details, including any co-interventions and descriptions of usual care comparators. |
| **Outcomes** | Outcome names, outcome definitions, outcome types (dichotomous or rate), units of analysis, adverse outcomes, person measuring/reporting, missing participants/data, reasons missing.<br>**_Qualitative studies_**<br>Data relating to the effects of the intervention from the perspectives of people with CKD only. Qualitative data relating to intervention acceptability or the perceptions of other groups (e.g., healthcare professionals) on the effects of the intervention will not be extracted. |

Reviewers will resolve disagreements by discussion, with recourse to a third reviewer if required. We will also contact study authors to verify key trial characteristics and obtain missing numerical outcome data where required.

## Quality Assessment in Individual Studies

**Quantitative studies.** We will use the NIH Study Quality Assessment Tools to assess the risk of bias for each study [32]. Each study will be classified using the following criteria: (a) "good" (the study is judged to be at the least risk of bias for all domains, and results are considered to be valid); (b) "fair" (the study is judged to be susceptible to some bias but deemed not sufficient to invalidate its results), and (c) "poor" (the study is judged to be at significant risk of bias in a way that substantially lowers confidence in the results). Each risk of bias judgement will be accompanied by a clear rationale, allowing transparency and future replication of the review process. This documentation will be made available in the supplementary materials to provide context for each decision.

To ensure the validity and reliability of our findings, two independent reviewers will assess the risk of bias for each included quantitative study. Any discrepancies between the two reviewers over the risk of bias will be resolved with the inclusion of a third reviewer. Calibration exercises will be conducted before the review to ensure consistency between reviewers in applying the risk of bias tools.

**Qualitative studies.** We will use the Joanna Briggs Institute (JBI) Critical Appraisal Checklist for Qualitative Research to assess the methodological strengths and limitations of the qualitative studies included [33]. The JBI tool is comprised of ten items and is widely used and has evaluates most concepts relating to methodological rigour in qualitative research [34]. The items within the checklist will not be scored, and the checklist will not be used to exclude studies based on quality. Instead, we will provide a narrative summary of studies by item to facilitate an overall assessment of methodological rigour and quality. As for the quantitative studies, two independent reviewers will apply the JBI tool for each included qualitative study, with recourse to a third in the event of any disagreements.

Mixed methods studies will be subject to both the NIH and the JBI tool for their quantitative and qualitative components.

## Data synthesis

**Quantitative studies.** Where appropriate, data from randomised controlled trials using comparable outcomes will be pooled in meta-analyses with a Frequentist framework. If there is a limited scope to undertake a meta-analysis, due to a small amount of existing trials with a range of different outcomes measured reported, we will create narrative synthesis of the findings using published guidance on synthesis without meta-analysis [35]. Data from non-randomised and observational studies will be pooled and synthesised separately in an exploratory analysis.

Data analyses will be conducted in Stata (StataCorp. 2023. Stata Statistical Software: Release 18. College Station, TX: StataCorp LLC.). Where the same measure is used to assess the same outcome, intervention effects will be summarised as mean differences. Where different measures are used to assess the same outcome, intervention effects will be summarised as standardised mean differences (SMD) (using Hedge's adjusted g) for the primary outcome measure.

If studies are sufficiently homogeneous in design, comparators, and outcome measures, we will conduct pairwise and network meta-analyses to estimate the relative intervention effects and their corresponding 95% confidence intervals. We will first conduct pairwise meta-analyses using a generic inverse variance random-effects model to compare individual interventions against control.

If there are sufficient included trials to perform network meta-analysis, we will then conduct a random-effects network meta-analyses to identify the most efficacious pharmacological and non-pharmacological interventions for fatigue management. Interventions that combine multiple approaches will be classified as distinct interventions within the analysis. Multi-arm studies will be accounted for by reweighting all comparisons of each multi-arm study [36]. Results from the network-meta-analyses will be presented as summary relative effects (MD or SMD) for every pair of interventions. Ranking probabilities for all interventions of being at each possible rank for each outcome will be estimated. We will obtain an intervention hierarchy using surface under the cumulative ranking curve (SUCRA) and mean ranks [37]. Inconsistency models will be used to evaluate the assumption of global consistency across the network. The separating indirect from direct evidence (SIDE) approach will be used to assess local consistency [37].

The validity of indirect comparisons relies on the transitivity assumption. Although variations in study populations (CKD stage) and intervention characteristics are anticipated, the included treatments will share similar therapeutic objectives. Therefore, it is considered plausible that relative treatment effects remain comparable across studies.

Important clinical and methodological characteristics that may be effect modifiers (including but not limited to demographics, CKD stage, intervention intensity, and study design) will be examined across comparisons. Where substantial heterogeneity is identified, subgroup analyses or meta-regression will be conducted where feasible. If, after these explorations, unexplained heterogeneity remains that undermines the assumptions of consistency and transitivity, the NMA results will not be reported. Transitivity will be assessed by comparing the distribution of effect modifiers across treatment comparisons. In cases of disconnected networks, analyses will focus on the largest connected network or be conducted separately for each network. Results will be interpreted with appropriate caution.

**Unit of analysis.** The unit of analysis will be the participant. If cluster randomised trials are identified, then analysis will be at the level of the individual while accounting for the clustering in the data (using the "design effect" in line with the

Cochrane Handbook). To calculate the design effect, the following data will be used: Number of cluster per intervention group; total number of participants per intervention group; outcome data (e.g., number of people who complete) and an estimate of the intracluster correlation coefficient (ICC). If the ICC is not reported, then it will be estimated from similar trials.

**Assessment of heterogeneity.** To evaluate the presence of clinical heterogeneity, descriptive statistics for study and study population characteristics will be compared across the included studies within each pairwise comparison.

For the pairwise meta-analyses, statistical heterogeneity will be quantified using the $I^2$ statistic. $I^2$ values will be interpreted alongside the estimates of the between-study variance (tau-squared). For network meta-analysis, the model will assume a common estimate for the between-study variance (tau-squared).

If heterogeneity is identified, potential causes will be explored (e.g., clinical and/or methodological diversity) through subgroup analyses or meta-regression for pre-specified variables. Potential sources of heterogeneity that will be considered include population characteristics (e.g.,CKD stage, renal replacement therapy status, co-morbidities), treatment characteristics, study design or risk of bias (allocation concealment, blinding, attrition), duration of follow-up, date of publication.

**Analysis of subgroups.** If possible, two subgroup analyses will be performed, one to explore the efficacy of interventions in different CKD cohorts (e.g., HD, PD, renal transplantation, CKM, and post-dialysis withdrawal) and the other to explore the efficacy of interventions on different types of fatigue – for example are some interventions more effective at addressing physical fatigue rather than mental or cognitive fatigue.

**Sensitivity analysis.** We will conduct sensitivity analyses to explore the impact of bias by excluding studies judged to be at high risk of bias. To evaluate the influence of measurement heterogeneity, a further sensitivity analyses will be conducted by stratifying studies according to instrument type:

- Fatigue-specific validated instruments

- Fatigue subscales from multidimensional instruments

**Assessment of publication bias.** If we can pool more than 10 studies in a meta-analysis, we will create and examine a funnel plot to explore possible small-trial and publication biases. Egger's test will also be employed to quantify the bias.

**Confidence in cumulative estimates.** Two reviewers will independently undertake an assessment of the certainty and confidence in evidence from the quantitative syntheses of the primary outcome, with recourse to a third to resolve any discrepancies as needed [38].

We will use Cochrane's GRADE approach (Grading of Recommendations Assessment, Development and Evaluation) [38] to judge the available evidence and categorise them into four levels of certainty by outcome: High, moderate, low, and very low. Five domains will be considered during the assessment process, which include: Risk of bias, inconsistency, indirectness, imprecision, and publication bias. Assessment of indirectness will include consideration of the representativeness of the study samples from the perspective of equality, diversity and inclusion.

**Qualitative evidence synthesis.** To synthesise qualitative findings, we will use the approach described by Thomas and Harden et al. (2008) [29]. Briefly, synthesis will follow three iterative stages: 1. line-by-line coding of the findings of primary studies; 2. The construction of descriptive themes which will be used to organise these initial codes and; 3. the development of 'analytical' themes which go beyond the descriptive themes to specifically address the review questions. This final stage will be conducted independently by two reviewers, before discussing, revising and agreeing these analytical themes, with recourse to the wider review team as required [29]. The synthesis process will be facilitated by NVivo (QSR International Ltd 2022, Version 20).

**Integration of quantitative evidence and qualitative evidence.** To integrate quantitative and qualitative evidence we will adopt a convergent segregated approach [39]. The separate quantitative and qualitative syntheses will be combined into a second level synthesis. Because the qualitative data extraction focuses on the perceived effectiveness

of interventions which address fatigue from the perspective of people living with CKD, we anticipate that this integrated synthesis will only focus on results relating to the effectiveness of these treatments. A joint display will be used to combine the findings in a tabulated form. This display will be used to assess the ways in which the data sets agree (confirmed), complement (offered an expanded explanation) or contradict each other and outline how the qualitative results expand and contextualise the key results from the quantitative results [39,40].

**Review status and timelines.** Information sources were searched and results uploaded to Covidence for screening in July 2025. We anticipate that record screening will be completed in January 2025, and the results of the review are expected in July 2026.

## Discussion

Existing systematic reviews focus upon limited interventions for fatigue, in specific CKD populations, highlighting the need for an updated systematic review to consolidate existing evidence in a broader range of interventions relevant across the CKD spectrum. Such a review is essential to guide the renal community in the effective, holistic management of fatigue and will contribute to clinical guidance in this area, as well as highlighting priority areas for future research in this area.

## Supporting information

**S1 Table. PRISMA-P Reporting Checklist.**
(DOCX)

**S2 Table. Draft Medline search strategy.**
(DOCX)

## Acknowledgments

The authors wish to thank Laura Lew and Bethan Pettifer for their support in developing this manuscript.

## Author contributions

**Conceptualization:** Hannah Mary Louise Young, Lisa Ancliffe, Ffion Curtis, Oluwafemi Efuntoye, Matthew P. M. Graham-Brown, Kathrine Parker, Chris Rolfe, Ellesha Smith, Aniebiot-Abasi Udofia, James O. Burton, Caroline Booth, Ishida Yuko.

**Methodology:** Hannah Mary Louise Young, Lisa Ancliffe, Ffion Curtis, Oluwafemi Efuntoye, Matthew P. M. Graham-Brown, Selina Lock, Kathrine Parker, Chris Rolfe, Ellesha Smith, Aniebiot-Abasi Udofia, Daniel Scott March.

**Supervision:** Daniel Scott March.

**Writing – original draft:** Hannah Mary Louise Young.

**Writing – review & editing:** Hannah Mary Louise Young, Lisa Ancliffe, Ffion Curtis, Oluwafemi Efuntoye, Matthew P. M. Graham-Brown, Selina Lock, Kathrine Parker, Chris Rolfe, Ellesha Smith, Aniebiot-Abasi Udofia, James O. Burton, Caroline Booth, Ishida Yuko, Daniel Scott March.

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
