## [Decision Letter · Decision Letter 0]

2 Mar 2026

PONE-D-25-55536The effectiveness of pharmacological and non-pharmacological interventions for fatigue in people living with chronic kidney disease: A protocol for a systematic review.PLOS One

Dear Dr. Young,

Thank you for submitting your manuscript to PLOS ONE. After careful consideration, we feel that it has merit but does not fully meet PLOS ONE’s publication criteria as it currently stands. Therefore, we invite you to submit a revised version of the manuscript that addresses the points raised during the review process.

**The protocol is well written and the research question relevant but I agree with the two reviewers that the populations of interest are many despite they are all under CKD. I recommend avoiding the outcome inclusion within the search of keywords in order not to miss papers. Please provide a point-by-point response to both reviewers.**

We look forward to receiving your revised manuscript.

Kind regards,

Mabel Aoun, MD, MPH

Academic Editor

PLOS One

**Journal Requirements:**

1. When submitting your revision, we need you to address these additional requirements. Please ensure that your manuscript meets PLOS ONE's style requirements, including those for file naming. The PLOS ONE style templates can be found at https://journals.plos.org/plosone/s/file?id=wjVg/PLOSOne_formatting_sample_main_body.pdf and https://journals.plos.org/plosone/s/file?id=ba62/PLOSOne_formatting_sample_title_authors_affiliations.pdf 2. Please provide a complete Data Availability Statement in the submission form, ensuring you include all necessary access information or a reason for why you are unable to make your data freely accessible. If your research concerns only data provided within your submission, please write "All data are in the manuscript and/or supporting information files" OR If no pilot data are reported in the manuscript, “No datasets were generated or analysed during the current study. All relevant data from this study will be made available upon study completion” as your Data Availability Statement. 3. Please include captions for your Supporting Information files at the end of your manuscript, and update any in-text citations to match accordingly. Please see our Supporting Information guidelines for more information: http://journals.plos.org/plosone/s/supporting-information. 4. If the reviewer comments include a recommendation to cite specific previously published works, please review and evaluate these publications to determine whether they are relevant and should be cited. There is no requirement to cite these works unless the editor has indicated otherwise.

Reviewers' comments:

Reviewer's Responses to Questions

**Comments to the Author**

1. Does the manuscript provide a valid rationale for the proposed study, with clearly identified and justified research questions?

Reviewer #1: Yes

Reviewer #2: Yes

2. Is the protocol technically sound and planned in a manner that will lead to a meaningful outcome and allow testing the stated hypotheses?

Reviewer #1: Partly

Reviewer #2: Partly

3. Is the methodology feasible and described in sufficient detail to allow the work to be replicable?

Reviewer #1: No

Reviewer #2: No

4. Have the authors described where all data underlying the findings will be made available when the study is complete?

Reviewer #1: No

Reviewer #2: Yes

5. Is the manuscript presented in an intelligible fashion and written in standard English?

Reviewer #1: Yes

Reviewer #2: Yes

6. Review Comments to the Author

You may also provide optional suggestions and comments to authors that they might find helpful in planning their study.

**Reviewer #1:** I read with great interest the protocol of the systematic review and meta-analysis (SRMA) evaluating the effectiveness of pharmacological and non-pharmacological interventions for fatigue in people living with chronic kidney disease (CKD).

The scope of this project is very broad and heterogeneous. The authors plan to assess all interventions targeting fatigue across all stages of CKD (eGFR < 60 mL/min/1.73 m²), including randomized controlled trials, non-randomized trials, and prospective studies, and additionally to conduct a qualitative systematic review of patient perspectives. While this ambition is commendable, the research question may be too wide and complex to address adequately within a single SRMA. A more focused scope—or clearer justification for combining such diverse interventions, populations, and study designs—would strengthen the protocol.

In addition, the introduction is overly long and would benefit from substantial condensation. It should be shortened to approximately one to one and a half pages, focusing more concisely on the rationale, existing gaps in the literature, and the specific objectives of the review.

I have two further comments:

1. Search strategy: The authors did not provide a sufficiently detailed description of the keywords and search terms that will be used. Including the full search strategy for at least one database would improve transparency and reproducibility.

2. Timeline inconsistency: The protocol was registered on PROSPERO in September 2025, yet the authors state that results will be available in July 2025. This appears to be a chronological error and should be clarified.

**Reviewer #2:** This protocol addresses an important and clinically relevant topic: the management of fatigue in individuals living with chronic kidney disease (CKD).

The proposed review is ambitious in scope, encompassing pharmacological and non-pharmacological interventions across the full CKD spectrum and integrating both quantitative and qualitative evidence. The protocol is generally well structured, registered in PROSPERO, and reports adherence to PRISMA-P guidance.

However, several methodological issues require clarification and refinement to ensure internal validity, feasibility, and interpretability of the planned analyses.

I recommend major revision prior to consideration for publication.

1. Clinical Heterogeneity

Multiple patient eligibility criteria (all CKD stages, HD, PD, transplanted patients…) and both pharmacological and non-pharmacological interventions are eligible, across multiple study designs (RCTs, non-randomised studies, before-after studies, qualitative studies).

The feasibility of synthesising such diverse populations and interventions is uncertain.

The protocol should:

- Provide a clearer justification of the transitivity assumption underpinning the proposed NMA.

- Clarify whether analyses will be stratified a priori by CKD population (e.g., dialysis vs non-dialysis vs transplant).

- Define conditions under which NMA would be deemed inappropriate due to heterogeneity.

2. Definition and Measurement of Fatigue

The protocol adopts a broad definition of fatigue. While this enhances inclusivity, it raises conceptual and statistical concerns: fatigue-specific instruments are not equivalent to fatigue subscales and different instruments may measure distinct fatigue dimensions.

The authors should:

- Define a hierarchy of outcome measures (e.g., fatigue-specific validated instruments as primary).

- Pre-specify how multidimensional fatigue instruments will be handled.

- Consider sensitivity analyses by instrument type.

3. Network Meta-Analysis: Feasibility and Assumptions

The planned NMA includes potentially diverse interventions

The protocol should clarify:

- Whether a frequentist or Bayesian NMA framework will be used.

- How multi-arm trials will be handled.

- How combination interventions will be classified.

4. Chronological Inconsistency

The protocol states that « Searches were conducted in July 2025. Screening anticipated completion in January 2025. Results expected July 2025. »

This timeline is internally inconsistent and must be corrected.

7. PLOS authors have the option to publish the peer review history of their article (what does this mean?). If published, this will include your full peer review and any attached files.

Reviewer #1: No

Reviewer #2: No

---

## [Author Response · Author response to Decision Letter 1]

17 Mar 2026

Please see the attached file for responses to the reviewers

---

## [Decision Letter · Decision Letter 1]

20 Apr 2026

The effectiveness of pharmacological and non-pharmacological interventions for fatigue in people living with chronic kidney disease: A protocol for a systematic review.

PONE-D-25-55536R1

Dear Dr. Young,

We’re pleased to inform you that your manuscript has been judged scientifically suitable for publication and will be formally accepted for publication once it meets all outstanding technical requirements.

Kind regards,

Mabel Aoun, MD, MPH

Academic Editor

PLOS One

Additional Editor Comments (optional):

Reviewers' comments:

Reviewer's Responses to Questions

**Comments to the Author**

1. Does the manuscript provide a valid rationale for the proposed study, with clearly identified and justified research questions?

Reviewer #1: Yes

Reviewer #2: Yes

2. Is the protocol technically sound and planned in a manner that will lead to a meaningful outcome and allow testing the stated hypotheses?

Reviewer #1: Yes

Reviewer #2: Yes

3. Is the methodology feasible and described in sufficient detail to allow the work to be replicable?

Reviewer #1: Yes

Reviewer #2: Yes

4. Have the authors described where all data underlying the findings will be made available when the study is complete?

Reviewer #1: Yes

Reviewer #2: Yes

5. Is the manuscript presented in an intelligible fashion and written in standard English?

Reviewer #1: Yes

Reviewer #2: Yes

6. Review Comments to the Author

You may also provide optional suggestions and comments to authors that they might find helpful in planning their study.

Reviewer #1: Thank you for addressing all the comments. The manuscript has improved and is now clearer for readers.

Reviewer #2: Thank you to the authors who responded to all the comments.

I think that the protocol is now ready to be published.

7. PLOS authors have the option to publish the peer review history of their article (what does this mean?). If published, this will include your full peer review and any attached files.

Reviewer #1: No

Reviewer #2: **Yes:** Ahmad MROUE

---

## [Editor Report · Acceptance letter]

PONE-D-25-55536R1

PLOS One

Dear Dr. Young,

I'm pleased to inform you that your manuscript has been deemed suitable for publication in PLOS One. Congratulations! Your manuscript is now being handed over to our production team.

Kind regards,

on behalf of

Pr Mabel Aoun

Academic Editor

PLOS One